# A Systematic Review of Government-Led Free Caesarean Section Policies in Low- and Middle-Income Countries from 2009 to 2025

**DOI:** 10.3390/healthcare13192522

**Published:** 2025-10-04

**Authors:** Victor Abiola Adepoju, Abdulrakib Abdulrahim, Qorinah Estiningtyas Sakilah Adnani

**Affiliations:** 1Department of HIV and Infectious Diseases, Jhpiego, an Affiliate of Johns Hopkins University, Abuja 900918, Nigeria; 2Department of Medical Microbiology, Faculty of Medicine and Health Sciences, Universiti Putra Malaysia, Serdang 43400, Malaysia; abdulrakib161@gmail.com; 3Department of Public Health, Faculty of Medicine, Universitas Padjadjaran, Bandung 40161, Indonesia; qorinah.adnani@unpad.ac.id

**Keywords:** Caesarean section, user-fee exemption, maternal health financing, low- and middle-income countries, health policy implementation, universal health coverage, out-of-pocket expenditure, service-use equity, provider payment reform, systematic review

## Abstract

**Background**: Caesarean section (CS) is a critical intervention, yet stark inequities in access persist across low- and middle-income countries (LMICs). Over the last decade, governments have introduced policies to eliminate or subsidize user fees; however, the collective impact of these initiatives on utilization, equity, and financial protection has not been fully synthesized. **Methods**: We conducted a systematic review in line with PRISMA 2020 guidelines. Searches were conducted in PubMed, Dimensions, Google Scholar, Scopus, Web of Science, and government portals for studies published between 1 January 2009 and 30 May 2025. Eligible studies evaluated government-initiated financing reforms, including full user-fee exemptions, partial subsidies, vouchers, insurance schemes, and provider-payment restructuring. Two reviewers independently applied the PICOS criteria, extracted data using a 15-item template, and assessed the study quality. Given heterogeneity, results were synthesized narratively. **Results**: Thirty-seven studies from 28 LMICs were included. Most (70%) evaluated fee exemptions. Mixed-methods and cross-sectional designs predominated, while only six studies employed interrupted time series designs. Twenty-two evaluations (59%) reported increased CS uptake, ranging from a 1.4-fold rise in Senegal to a threefold increase in Kano State, Nigeria. Similar surges were also observed in non-African contexts such as Iran and Georgia, where reforms included incentives for vaginal delivery or punitive tariffs to curb overuse. Fourteen of 26 fee-exemption studies documented pro-rich or pro-urban drift, while catastrophic expenditure persisted for 12–43% of households, despite the implementation of “free” policies. Median out-of-pocket costs ranged from USD 14 in Burkina Faso to nearly USD 300 in Dakar’s slums. Only one study linked reforms to a reduction in neonatal mortality (a 30% decrease in Mali/Benin), while none demonstrated an impact on maternal mortality. Qualitative evidence highlighted hidden costs, delayed reimbursements, and weak accountability. At the same time, China and Bangladesh demonstrated how demographic reforms or voucher schemes could inadvertently lead to CS overuse or expose gaps in service readiness. **Conclusions**: Government-led financing reforms consistently increased CS volumes but fell short of ensuring equity, financial protection, or sustained quality. Effective initiatives combined fee removal with investments in surgical capacity, timely reimbursement, and transparent accountability. Future CS policies must integrate real-time monitoring of equity and quality and adopt robust quasi-experimental designs to enable mid-course correction.

## 1. Introduction

Caesarean section (CS) remains one of the most decisive innovations in modern obstetrics. Timely access to surgery can prevent roughly one in five maternal deaths and a comparable share of intrapartum stillbirths [1]. Yet access to CS is profoundly inequitable. In 2021, the median CS rate in low-income countries was only 5%, far below the 10–15% range considered necessary to address major obstetric complications. By contrast, upper-middle-income countries reported median rates of 36%, with several exceeding 50% [2,3,4,5]. Within many LMICs, this disparity deepens as urban elites undergo CS at rates comparable to high-income settings. At the same time, rural and poorer women remain excluded from timely anesthesia, blood transfusion, and safe operating theatres [6]. Closing this gap is central to United Nations Sustainable Development Goal 3, which commits governments to end preventable maternal and neonatal deaths by 2030.

Three barriers perpetuate inequity. First, the cost of surgery at the point of care remains prohibitive. In West Africa, families historically paid USD 100–300 for an emergency CS, which is equivalent to several months of household income [7]. Second, surgical capacity is highly uneven. The Lancet Commission on Global Surgery estimated that only 6% of hospitals in sub-Saharan Africa meet minimum standards for safe abdominal delivery [8]. Third, weak accountability ecosystems allow informal charges, unclear entitlements, and delayed reimbursements to undermine trust [9].

In response, governments have introduced reforms, such as those eliminating or subsidizing user fees for maternity and CS services. Over 40 LMICs have now adopted such measures, spanning sub-Saharan Africa, South Asia, the Middle East, and parts of Eastern Europe. These reforms rest on a strong economic logic that removing user fees reduces immediate financial barriers, encourages earlier care-seeking, and should, in theory, improve equity by benefitting poorer households disproportionately affected [10]. In practice, however, implementation has been uneven. Reports from Senegal, Mali, and Kenya documented rapid surges in CS following fee removal but also persistent stock-outs, high transport costs, and urban concentration of benefits [11,12]. Conversely, Iran’s Health Sector Evolution Plan and Georgia’s punitive tariff system illustrate that reforms can also reduce or reconfigure the utilization of CS, though often with mixed consequences for equity and quality [13,14]. Throughout this paper, equity is referred to as a programmatic goal. Equity can be defined as just distribution of resources in society.

Evidence to date is fragmented. Most evaluations rely on simple pre–post designs, limiting causal inference and insight into distributional effects. Related reviews have assessed broader financing reforms (such as health insurance or regulatory incentives) [15], but none have synthesized how government-led interventions specifically targeting access to CS have shaped utilization, equity, financial protection, and quality across LMICs. Non-African contexts, including Bangladesh’s maternal voucher scheme [16,17] and China’s universal two-child policy [18]. They remain underrepresented in global syntheses despite their relevance.

This review addresses that gap. We systematically collated and appraised studies evaluating government-led reforms that include full fee exemption, partial subsidies, vouchers, insurance schemes, and provider payment redesigns aimed at improving access to CS in LMICs. We focused exclusively on government-initiated efforts that are scalable at a national level, rather than small-scale NGO pilots. Our analysis extends beyond utilization to examine equity gradients, out-of-pocket expenditures, clinical outcomes, and implementation fidelity. We synthesized evidence across Africa, Asia, the Middle East, and Eastern Europe to support ministries confronting fiscal pressures and policy trade-offs. This synthesis is timely as many LMICs navigate post-COVID budgetary constraints and policy resets that accompanied the 2025 dip in donor funds from the United States. Hence, we aggregate evidence up to May 2025 to inform near-term financing choices. Conceptually, we view fee removal policies acting through three pathways, namely: (i) price signal to demand (utilization); (ii) supply readiness & provider incentives to quality/appropriate use; and (iii) accountability & reimbursement timelines to financial protection/equity. We use this framework to organize the synthesis. Our ultimate objective is to guide governments toward reforms that expand access to life-saving CS while ensuring equity, financial protection, and judicious use.

## 2. Method

### 2.1. Study Design

Three reviewers (VAA, AA, QESA) conducted a systematic review in full compliance with the PRISMA 2020 statement. The protocol was registered on the Open Science Framework (OSF; Center for Open Science, Charlottesville, VA, USA; registration number: https://doi.org/10.17605/OSF.IO/5VN7D) on 3 August 2025, to ensure transparency of methods and decisions, given that the original research questions evolved from an ongoing policy dialogue.

### 2.2. Eligibility Criteria

We applied the PICOS framework (Population, Intervention, Comparison, Outcomes, Study type) to guide inclusion and exclusion (Table 1). Studies were eligible if they evaluated government-led or government-initiated policies that removed or altered user charges for CS or delivery care, including full exemptions, partial subsidies, insurance packages, vouchers, provider payment reforms, or bundled sector-wide reforms with a financing component. Broader sector-wide reform bundles are multi-component, system-level packages that go beyond fee exemptions and combine changes to provider payment, benefit packages, human resources, infrastructure & supplies/medicines, organization of service delivery organization, and governance/data systems, all implemented together as a health-system reform bundle with CS/maternal services embedded, not a stand-alone user-fee policy. Eligible designs included quantitative, qualitative, and mixed-methods primary research conducted in World Bank–classified LMICs from 1 January 2009 to 30 May 2025. We excluded purely private sector schemes, NGO-led pilots without government scale-up, clinical trials without policy content, non-English studies, commentaries, and reviews without primary data.

### 2.3. Information Sources and Search Strategy

We systematically searched PubMed, Scopus, Dimensions, Web of Science, and Google Scholar from 1 January 2009 to 30 May 2025. We also explored the WHO, World Bank, UNICEF, and national government portals for unpublished technical reports, and scanned the reference lists of prior reviews. Search strings combined terms for financing, CS, and LMICs, tailored to each database. The complete Boolean queries, search interfaces, and dates of final runs are provided in Appendix A for reproducibility.

### 2.4. Selection Process

Database searches yielded 2808 records across five sources: PubMed (*n* = 21), Scopus (*n* = 131), Dimensions (*n* = 2), Web of Science (*n* = 34), and Google Scholar (*n* = 2620). After deduplication and removal of obvious non-research items, 2354 unique records proceeded to title/abstract screening. Two reviewers (AA and QESA) independently applied the pre-specified eligibility criteria at both screening stages. Of these, 1656 records were excluded at the title stage as clearly out of scope for LMIC, government-led CS/financing content. We then assessed 698 abstracts in detail and retrieved 131 full texts. Ninety-four full texts were excluded, with reasons disaggregated as follows: no caesarean government financing component (*n* = 34), no CS-related outcome (*n* = 28), editorial/commentary with no primary data (*n* = 19), and non-English or inaccessible full text (*n* = 13). The inter-reviewer agreement at the full-text stage was 92%, and disagreements were resolved by consensus. Thirty-seven studies met the inclusion criteria and were synthesized. The PRISMA flow diagram (Figure 1) reflects these tallies and the disaggregated exclusion categories. A complete list of excluded full texts, along with the reasons for exclusion, is provided in Appendix A.

### 2.5. Data Collection Process

We developed a structured 15-item, excel-based extraction template that covers policy features, funding sources, design, outcomes, effect sizes (where reported), and implementation dynamics. One reviewer (VAA) extracted data, and a second reviewer (AA) independently cross-checked all entries. No automation tools were used. When details were unclear, we consulted the appendices or contacted the study authors.

### 2.6. Data Items

Primary outcomes were CS utilization, facility delivery, and maternal/neonatal outcomes (mortality, morbidity, complications). Secondary outcomes were equity gradients (wealth, education, residence), financial protection (out-of-pocket costs, catastrophic expenditure), and implementation processes (reimbursement timeliness, accountability, staff workload). We captured quantitative effect descriptions where available and qualitative explanatory themes.

### 2.7. Risk of Bias Assessment

We assessed methodological credibility for all 37 included studies and did not exclude based on quality. For quantitative observational/quasi-experimental designs, we used ROBINS-I; for interrupted time series, we used the Cochrane EPOC ITS tool; and for qualitative/realist/process evaluations, we used the CASP qualitative checklist. Two reviewers appraised each study independently and resolved disagreements by consensus (see the “RoB consensus” column in Appendix A). For mixed-methods articles, we assigned a single primary appraisal tool according to the dominant evaluation component: impact (ROBINS-I), ITS (EPOC-ITS), or qualitative/process (CASP). To avoid double-counting, synthesis counts, therefore, refer to studies, not study–method rows. Per-study judgements are provided in Appendix A, and overall traffic-light summaries by tool are shown in Appendix A (CASP), Appendix A (EPOC-ITS), and Appendix A (ROBINS-I).

### 2.8. Methods of Synthesis

Given substantial heterogeneity in interventions and designs, we did not conduct a meta-analysis. We conducted a narrative synthesis structured around (1) policy design, (2) utilization/coverage, (3) equity and financial protection, and (4) implementation dynamics. We compared direction and approximate magnitude effects without transformation, explored apparent contradictions by policy design, setting, and implementation fidelity, and privileged higher credibility designs in the text.

### 2.9. Reporting Bias and Certainty Assessment

We explored possible reporting bias by comparing included studies with excluded records and by checking for selective outcome reporting against study protocols where available. Funnel plot asymmetry was not assessed due to heterogeneity and small sample sizes within subgroups. Although we did not formally apply GRADE, we qualitatively assessed the certainty of the body of evidence and highlighted limitations related to confounding, routine data, short follow-up, and heterogeneity.

## 3. Results

### 3.1. Characteristics of Included Studies

This review synthesized evidence from 37 empirical studies of government-led policies aimed at expanding access to CS and related maternity care in LMICs (See Table 2; studies listed by intervention category). The evidence base has accumulated steadily since 2009, with a marked peak between 2016 and 2018 when 13 studies (35%) were published [7,9,13,14,19,20,21,22,23,24,25,26,27]. Publication slowed somewhat thereafter, with nine studies (24%) appearing between 2019 and 2021 [24,28,29,30,31,32,33,34,35] and four (11%) between 2022 and 2024 [18,36,37,38].

Geographically, the evidence remains dominated by Sub-Saharan Africa, which accounted for 27 of the 37 studies (73%) [7,11,12,13,19,20,21,23,24,26,28,29,30,31,32,33,34,35,37,39,40]. Three studies originated from the Middle East and North Africa [14,41,42], one from Georgia at the intersection of Eastern Europe and West Asia [36], and one from China [18]. Fifteen studies were conducted in low-income countries [7,11,13,20,21,24,27,28,31,34,37,43,44,45], another 15 in lower-middle-income countries [12,20,21,24,25,26,29,30,31,32,33,34,37,39,41,42,43,44,46,47], and seven in upper-middle-income settings [14,18,22,36,42].

Most reforms were framed around exempting user fees. Twenty-six studies (70%) examined policies that entirely removed patient charges for CS [7,11,12,13,20,21,24,25,26,28,29,30,31,32,33,34,35,37,39,41,42,43,44,45,46]. In comparison, four studies (11%) evaluated partial subsidies [13,21,45,48], three (8%) assessed voucher or demand-side financing mechanisms [16,17,46], and three (8%) analyzed broader sector-wide reform bundles [13,22,28]. A single study assessed a punitive reimbursement model that imposed penalties for overuse of CS [36].

The study designs were heterogeneous. Ten employed cross-sectional approaches (27%) [27,29,31,32,37,42,43,47,48], while 13 adopted mixed-methods designs (35%) [7,13,21,23,24,31,34,40,41,46]. Interrupted time series analyses were used in six studies (16%) [11,14,28,30,33,36], three applied quasi-experimental difference-in-differences or regression-discontinuity approaches (8%) [26,27,46], and three were qualitative (8%) [16,19,34]. One used realist evaluation (3%) [25], and one was an economic case–control analysis (3%) [47]. No randomized controlled trials were identified. Annual publication volume is shown in Figure 2, and the distribution of policy instruments in Figure 3.
healthcare-13-02522-t002_Table 2Table 2Characteristics of included studies.Serial/NumberAuthor, Year CountryWorld Bank Income Category Title of PolicyType of Gov’t InterventionGov’t Led?Policy AimType of EvaluationOutcomes MeasuredResults SummaryLimitations ReportedSummary of Quantitative and Qualitative FindingsReflection on Policy Implementation Weaknesses/StrengthsEligible?If Yes to Eligible—Reason1.Ravit et al. [27]Benin, MaliLow income (both)Free CS PolicyUser fee exemption for CSYesTo improve access to C-sections and reduce socioeconomic and geographic disparities in maternal healthcareObservational study using repeated cross-sectional DHS dataC-section rate, facility-based delivery, inequalities by wealth, education, and locationNo substantial reduction in socioeconomic inequalities; some improvement in education-related inequality in MaliSelf-reported data, excluding stillbirths, incomplete implementation of policy, and limited post-policy yearsQuantitative findings showed slight improvements in Mali; inequalities remained in both countries. No qualitative data presented.Strength: increased access to C-sections. Weakness: persistent wealth-related inequality; policy is insufficient alone; implementation gaps.YesGovernment-led, evaluated C-section and equity outcomes in LMICs, based on national policy2.Sylla et al. [37]SenegalLower middleFree Caesarean Section PolicyHealth financing reform/subsidyYesTo remove financial barriers to emergency Caesarean sections by providing free access in public health facilities.Quantitative cross-sectional retrospective studyAccess to C-section, out-of-pocket expenditure, financial burden, and implementation barriersMany low-income women still incur significant costs despite the policy; costs vary widely by location and facility, and delayed state reimbursements are often cited as the cause.Does not examine overuse or medically unnecessary CS; it excludes wealthier populations and those who cannot access services.240 slum-dwelling women underwent CS between July–December 2022. Despite the policy, the average cost was approximately $ 296 USD. 75% of households could not meet basic needs. Facility managers underestimated the total costs borne by patients.Strength: Policy ensures medical need is a priority in CS provision. Weakness: Delayed reimbursements, hidden costs, and poor enforcement lead to significant patient payments and undermine policy goals.YesFocuses on government-led free CS policy in LMIC with quantitative evaluation of health system outcomes.3.Witter et al. [41]SenegalLower middle incomeFree Delivery and Caesarean Policy (FDCP)Fee exemption for deliveries and CaesareansYesRemove financial barriers to facility-based deliveries and CS to reduce maternal mortalityMixed-methods: policy evaluation, costing, qualitative interviews, and clinical record analysisCS rates, facility delivery rates, financial impact, staff workload, equity, household costsCS rate rose from 4.2% to 5.6%; supervised deliveries rose from 40% to 44%; cost per additional CS = $467; financial and logistical challenges constrained impactNo national household survey; missing clinical data in some sites; unclear resource flow to lower-level facilities; equity impact limited in rural areasQuantitative: CS and delivery rates improved; financial transfers benefited regional hospitals more than health posts. Qualitative: persistent user costs, confusion over policy scope, and undercompensation of community health workers.Strengths: Increase in CS use and supervised deliveries; low per capita cost; political will. Weaknesses: Poor understanding at the facility and community level; delays and inequities in resource allocation; no compensation to health posts; users still pay for many items.YesGovernment-led, LMIC, original data, CS-focused, health system, and outcome metrics reported4.Cavallaro et al. [28]TanzaniaLow incomeHealth Sector Reform with a maternal health exemption componentHealth system reform and user fee exemptionYesImprove access to maternal services, including free delivery and CS, in public facilitiesInterrupted time series analysis using HMIS dataC-section rate, facility-based delivery rate, and regional equityThe national CS rate increased from 2.5% to 4.8% after the policy was implemented. The most significant increases were seen in urban hospitals. The rural-urban gap widened slightly.No control group; cannot disaggregate medically necessary CS; weak data from some regionsCS rates increased notably in hospitals but were limited in rural clinics. Data analysis revealed that CS growth was primarily observed among educated, urban women.Strength: National rollout, improved access in tertiary centers. Weakness: Persistent regional inequities, rural service gaps, uneven implementation.YesGovernment-led evaluation of CS outcomes post-policy in LMIC, with time-trend analysis5.Arsenault et al. [7]MaliLowUser fee exemption policy for CSFee exemption for C-sections and referral support systemYesReduce maternal mortality and the economic burden of emergency obstetric care, including C-section costsQuantitative and qualitative analysis (cross-sectional, logistic regression, household surveys)Catastrophic expenditure, financial coping mechanisms, CS cost coverage, impoverishment effectsDespite fee exemption, 40–43% of C-section cases still faced catastrophic spending due to prescription drugs, transport, and out-of-pocket paymentsNo direct income data; results not generalizable to all of Mali or to women who did not reach the facilityC-section cost coverage failed to prevent catastrophic spending; CS patients paid for drugs outside kits and faced transportation costs. 43% incurred catastrophic costs. 44.6% reduced food, 23.2% still in debt, >10 months later.Strength: Fee exemption policy exists. Weaknesses: Incomplete kit coverage, poor stock management, transportation not fully covered, weak referral fund performance, and equity gaps remain.YesGovernment-led fee exemption for C-sections, original data, with outcomes on CS costs and impact6.Karami Matin et al. [14]IranUpper middleHealth Sector Evolution Plan (HSEP)Health financing and service reformYesReduce C-section rates and improve access to hospital servicesInterrupted Time Series AnalysisHospitalization rate, Caesarean section rateHospitalization increased significantly post-HSEP. The initial drop in C-section rate was not sustained; a rising trend followed in subsequent months.Focus on a single province, possible unmeasured confounders, and short post-intervention follow-up.Quantitative ITS showed a temporary decline in C-section rate but a long-term upward trend. Hospital utilization rose significantly post-policy.Strength: Immediate implementation impact on CS and access. Weakness: Lack of sustained CS reduction; trend reversal implies inadequate cultural or systemic support for vaginal delivery.YesGovernment-led reform, outcome data on CS, ITS design, LMIC context7.Nedberg et al. [36]GeorgiaUpper middleNational Caesarean Section (CS) Reduction PolicyPunitive financial penalties for non-compliance with CS rate reduction targetsYesTo reduce high national Caesarean section ratesInterrupted Time Series Analysis (ITSA)CS rate, NICU transfers, perinatal mortalityCS rate dropped from 44.7% to 40.8%, with the largest decrease among primiparous women; NICU and PM rates remained largely unchanged.Short follow-up period, inability to distinguish between medically indicated and elective CS, and rare PM events susceptible to fluctuationQuantitatively, significant CS rate reduction, especially among young and educated women; no adverse effect on NICU or PM rates reported. Qualitative interpretations are limited.Strength: Achieved CS reduction nationally using a unique punitive financial model. Weakness: Lack of stakeholder engagement and potential unintended consequences not assessed.YesGovernment-led policy in an LMIC with quantitative outcome data on CS and perinatal health8.Rooeintan et al. [22]IranUpper middleIran Healthcare Evolution PlanComprehensive healthcare reform including incentives for vaginal deliveryYesTo increase vaginal delivery and reduce Caesarean section ratesPre-post cross-sectional quantitative studyRates of vaginal delivery, Caesarean section, painless delivery, midwife-assisted deliverySignificant increase in vaginal delivery (35.3% to 41.4%); CS declined from 64.7% to 58.6% but was not statistically significant.No separation of primary/repeat CS, small sample for painless/midwife deliveries, limited facility participationQuantitatively, vaginal delivery increased significantly in public hospitals; qualitative observations suggest reform was ineffective in private hospitals and under-resourced facilities.Strength: Free vaginal delivery and policy focus increased uptake in public hospitals. Weakness: Limited reach to private hospitals, insufficient midwife infrastructure, and small painless delivery rollout.YesGovernment-led reform in an LMIC with pre-post CS outcome data9.Tang et al. [18]ChinaUpper middle incomeUniversal Two-Child PolicyNational fertility policy reformYesTo address demographic decline and an aging population by increasing fertilityRetrospective cross-sectional cohort analysisCaesarean section rate stratified by maternal age and Robson classificationCS rate rose with maternal age: 36.1% (20–34), 57.9% (35–39), 64.75% (¥40); group 5 contributed most (51.03%) to CS; AMA was strongly linked to higher CS rates.Single tertiary center, retrospective design, limited generalizability due to urban and mobile populationQuantitative: AMA is significantly associated with higher CS rates, especially in groups 1, 5, and 10. CS rate peaked during COVID-19. No qualitative data included.Strengths: Robust use of Robson classification, large sample size, policy-relevant insights. Weaknesses: Lacked qualitative data, limited generalizability, and no detailed analysis of individual CS decision factors.YesLMIC context, government-led policy, analysis of CS-related outcomes with stratified evaluation10.Orangi et al. [33]KenyaLower middle incomeFree Maternity Policy and Linda Mama ProgrammeUser fee removal; national insurance-based financing reformYesTo remove financial barriers and improve access to maternal health servicesInterrupted Time series (ITS) AnalysisNormal deliveries, CSs, antenatal care (ANC) visits, postnatal care (PNC) visitsThe 2013 policy led to a 19.6% and 28.9% increase in normal deliveries and CS in public facilities. Linda Mama had mixed effects: a trend decrease in CS in public, a level decrease in private sector CS, and normal deliveries.Missing data (27–55%), assumption of uniform implementation dates, lack of control for geographic/access barriers, inability to capture all cointerventionsQuantitative: CS and normal deliveries increased initially in public facilities. Trend effects waned over time. Mixed results in private/faith-based facilities. Qualitative: Not included.Strengths: National scale, robust ITS design, addresses real-world implementation. Weaknesses: Inadequate reimbursement, poor implementation fidelity, persistent supply-side issues, lack of clarity on benefit packages.YesGovernment-led policy in an LMIC, focused on CS rate and maternal health service use, with a robust evaluation11.Orangi et al. [35]KenyaLower middleLinda Mama Free Maternity ProgramHealth financing reform / free maternity policyYesImprove access, reduce inequities, enhance accountability, and extend maternal care services, including CS.Mixed-methods cross-sectional process evaluationOut-of-pocket payments, service coverage (including CS), facility revenue, reimbursement delaysPolicy expanded services include. CS; implementation gaps led to continued OOP expenses, low CS reimbursement, & service denialCross-sectional design, limited generalizability, exclusion of private-for-profit facilitiesDespite CS reimbursement being included (KES 17,000), many facilities reported CS not being covered in practice. Delays in fund disbursement and low rates limited provider participation. 45–52% of mothers incurred OOP costs for delivery, often in faith-based settings.Strengths: Expanded benefit package, intent to improve equity. Weaknesses: Poor communication, insufficient reimbursement rates (esp. for CS), reimbursement delays, facility-level financial barriers, and lack of essential commodities.YesGovernment-led policy with CS-specific content, evaluated using original data12.Lang’at et al. [30]KenyaLower middleFree Maternity Service (FMS) PolicyRemoval of user fees for maternity services in public facilitiesYesImprove access, use, and quality of maternity care services to reduce maternal and perinatal mortality.Interrupted time series analysis (quantitative, retrospective observational)ANC visits, health facility deliveries, live births, C-section rates, emergency obstetric care, stillbirth ratesSignificant increases in ANC visits (98%), deliveries (97%), and live births (89%). 27% rise in emergency obstetric care use. No significant change in C-section or stillbirth rates.No control group; only three counties studied; potential influence of unmeasured confounders; reliance on routine facility dataQuantitative analysis showed strong immediate and sustained improvements in utilization of skilled maternal care; limited impact on CS and stillbirth rates; ANC, deliveries, and emergency obstetric care use rose significantly.Strengths: strong uptake response shows affordability was a key barrier. Weaknesses: no observed improvement in CS/stillbirth rates, possibly due to unchanged facility capacity, staffing, or quality.YesGovernment-led policy in LMIC with outcome data on maternal care access and Caesarean section usage, using the ITS design13.Ahmed et al. [16]BangladeshLower middle incomeDemand-side Financing Maternal Health Voucher SchemeVoucher-based demand-side financingYesIncrease maternal health service utilization and reduce financial barriers to care, including CSQualitative evaluation through semi-structured stakeholder interviewsCS rate, facility delivery, ANC/PNC uptake, satisfaction, provider incentive structure43 CS supported; only 10 were performed locally due to a lack of an anesthesiologist; others were referredDelayed reimbursements, workforce shortages, and service readiness gapsIncreased ANC/delivery uptake; 43 CS supported, but referral needed due to inadequate facility capacity; positive beneficiary perceptionStrength: boosted utilization among the poor; Weakness: limited CS delivery capacity, unclear eligibility enforcement, funding delaysYesGovernment-led DSF scheme in an LMIC with direct focus on CS access, measurable outcomes, and qualitative evaluation14.Meda et al. [31]Burkina FasoLow incomeFree Maternal Health Care Policy Prospective fee-for-service financing with full subsidyYesEliminate OOP expenses for women during ANC, delivery, and EmONC, including C-sections.Cross-sectional survey using structured questionnaires at 299 health facilitiesCS-related OOP costs, delivery costs, drug availability, patient and facility characteristicsDespite the free care policy, 57.1% of women who had C-sections made OOP payments; the median CS-related cost was $13.78Stock-outs of drugs, urban-rural disparities, cleaning product costs, and some CS-related drugs are unavailable in facility pharmacies.29.6% of women made OOP payments, 57.1% of CS cases had payments; CS cost median = $136.39, OOP = $13.78; mostly for drugsStrengths: Nationwide reach, prospectively funded policy. Weaknesses: Drug stock-outs, regional disparities, CS burden persistsYesLMIC context, government-led national policy; evaluated CS outcomes with disaggregated analysis15.Ridde et al. [21]Burkina FasoLow incomeNational Subsidy for Deliveries and Emergency Obstetric and Neonatal Care (EmONC)Subsidy on delivery and Caesarean section costsYesTo reduce financial barriers and increase access to skilled delivery and emergency obstetric care, including CSsMixed-methods (qualitative interviews, focus groups, quantitative service data)Rate of assisted deliveries, financial protection, and implementation effectivenessAssisted deliveries increased post-policy; however, the trend began before the policy. The policy benefited women, health workers, and management committees, but indigent populations remained underserved.Lack of baseline data for impact assessment, no progressive rollout, poor definition of indigence criteria, weak communication, and evaluation gapsQuant: Rise in assisted deliveries; mixed effect on equity. Qual: Implementation gaps, variable interpretation of policy, lack of clarity on bonuses and indigent coverage, health worker incentives, and informal paymentsStrengths: National budget allocation, broad coverage, stakeholder participation, and transport inclusion. Weaknesses: Poor communication, insufficient funding for support activities, weak monitoring and evaluation, no indigent selection criteriaYesLMIC setting, government-led policy targeting CS use, with evaluation of CS-related outcomes16.Dossou et al. [24]BeninLow incomeUser Fee Exemption Policy for CSsFull cost exemption for CSs via fixed reimbursement to facilitiesYesTo end hospital detention of women/newborns due to unpaid user fees by removing financial barriers to CSMixed-methods (case study with quantitative and qualitative tools)CS rate, financial protection (fees paid as % of GDP per capita), equity of accessCS rates increased from 2.3% (2001) to 7% (2015). Policy reduced CS fees by 47–84%, but poor implementation and equity gaps remained. Richer women benefitted most.Policy not codified in law; weak monitoring; facility discretion led to user fees persisting; inconsistent definition of covered services; no strategic purchasing or quality assuranceQuant: CS rate rose 0.5% annually post-policy; costs dropped, but not eliminated. Equity worsened post-policy. Qual: Policy clashed with entrenched user fee culture; limited support from donors; governance and communication gapsStrengths: High-level political support; simple fixed reimbursement; reduced financial barriers for some. Weaknesses: Partial implementation; lack of provider guidelines; inequity in benefits; persistent fees; resistance due to pro-user fee cultureYesLMIC setting, government-led policy targeting CS rates with mixed-method evaluation of health and equity outcomes17.Ravit et al. [47]MaliLow incomeCS Fee Exemption Policy (2005)User fee exemption for CS in EmONC settingsYesTo eliminate financial barriers and improve access to emergency obstetric care, including CSQuantitative retrospective case–control study with economic analysisDirect/indirect costs, treatment-related expenses, equity of access91% of women still incurred significant treatment costs despite fee exemption; average expense was 77,017 FCFA (~163 USD), with treatment, transportation, and food costs highest among rural and poorer women.Recall bias, limited generalizability beyond Kayes region, over-representation of deceased women, diagnosis accuracy limits, proxy wealth measurement.Out-of-pocket spending remained high. Rural, poorer women spent more on transport and drugs. Near-miss cases had higher treatment costs. Systemic issues like unclear kit provisions and missing blood drove cost variability.Strengths: policy increased CS access; Weaknesses: high hidden costs, non-functional referral system, poor definition of fee exemption scope, inequality by wealth and residence, and insufficient supply of drugs/blood.YesFocused on CS policy in an LMIC with measured maternal outcomes and expenses18.Bennis et al. [42]MoroccoLower middle incomeFee exemption policy for delivery and CS in public hospitalsHealth financing reform (user fee removal)YesTo improve access to emergency obstetric care and reduce maternal mortalityCross-sectional observational study with interviews and cost analysisDirect household expenditure on CS, access, and hidden costsAverage cost reduced by 40% at SEGMA hospitals; UH still charged fees, creating inequities and high out-of-pocket costs.Short study period, limited generalizability beyond three hospitals, did not assess health outcomes or equity of benefit distributionHouseholds spent $169 on average in SEGMA and $291 in UH; hidden costs persisted despite formal fee removal. Coping strategies involved loans, handouts, or asset sales. Informal payments were common.Strength: Reduced average cost by 40%; Weakness: Inconsistent application (especially at UH), persistent hidden costs, and lack of full financial protection for the poorest.YesGovernment-led policy targeting CS with outcome data on CS-related costs in LMIC19.Khan et al. [49]PakistanLower middle incomePublic maternity care subsidy at tertiary government hospitalsSubsidized healthcare (no user fee waiver policy formally evaluated)YesTo subsidize delivery care at public hospitals and reduce the cost burden on householdsCross-sectional cost analysis (hospital and patient perspectives)Average cost of CS and SVD from both hospital and patient perspectivesThe average CS cost to patients was $204, and to hospitals, it was $162. Despite subsidies, patients bore 56% of the CS cost, making facility births unaffordable for most low-income households.Did not include indirect/intangible costs fully; shared service costs (lab/blood bank) lacked granularity; excluded private and rural facility dataPatients incurred $204 for CS, with drugs and hospital dues being the main cost drivers. 74% of SVD and 54% of CS households earned < $149/month. Costs exceeded affordability, causing financial strain.Strength: Provided critical baseline cost data for public CS/SVD services. Weakness: No formal fee exemption policy; poor families still faced high out-of-pocket costs; hidden costs remain substantial.YesCS-focused, LMIC, evaluates government-subsidized care delivery with cost outcomes20.McKinnon et al. [46]Ghana, Kenya, SenegalLower middle incomeDelivery Fee Exemption PolicyHealth financing reform (user fee removal)YesIncrease facility-based delivery and reduce neonatal mortality by removing user fees for delivery servicesDifference-in-differences (quasi-experimental)Facility-based deliveries, CSs, Neonatal mortality rate (NMR)Increased facility deliveries by 3.1 per 100 live births; no significant change in CS rates; potential reduction in neonatal mortality (−2.9/1000 births)Recall bias, variation in policy scope, differences in regional implementation, reliance on self-reported DHS dataQuantitatively, facility deliveries increased, and neonatal mortality possibly decreased; no increase in CS rates. No qualitative data reported.Strengths: Improved access to delivery services. Weaknesses: Policy is not sufficient to increase CS rates; other barriers like geographic access and infrastructure remain unaddressed.YesGovernment-led policy in LMICs with evaluation of CS and related outcomes using a valid study design21.Schantz et al. [34]Benin, MaliLow incomeFree CS PolicyUser fee removal policy for CSYesImprove access to emergency obstetric care and reduce maternal mortalityQualitative (interviews, observations, workshops)Facility CS rates, maternal experiences, provider behavior, systemic constraintsHigh facility CS rates (31% Mali, 43.9% Benin), including among low-risk women. Many CS performed for maternal distress or preventive reasons in under-resourced settings.Lack of generalizability, qualitative data limits causality, absence of national-level dataCS is often driven by maternal fear, pain, lack of privacy/support, inadequate staffing/equipment. Overuse is linked to weak supervision and funding incentives.Strength: Increased access to lifesaving CS. Weakness: Encouraged non-medically indicated CS due to suffering, staff burnout, and misuse of policy incentives.YesGovernment-led CS policy in LMICs with a focus on CS rates and non-medical drivers22.Ravit et al. [27]Mali and BeninLow incomeFree Caesarean PolicyUser fee exemption for CSYesTo increase access to CSs and facility-based deliveries, and reduce neonatal mortality by removing financial barriersQuasi-experimental (difference-in-differences using DHS data)CS rate, facility-based delivery rate, and neonatal mortalityCS increased by 36% overall; the strongest effects were for non-educated, rural, and mid-income women. FBD and hospital-based deliveries also increased. Neonatal mortality decreased by 30%.Quasi-experimental design limits causal inference; differences in policy implementation across sites; stillbirths excluded; short post-policy period for Benin.Quantitatively, CS, FBD, and hospital deliveries rose significantly. Neonatal deaths declined. Qualitative observations suggest poor implementation, incomplete kits, and residual user costs in practice.Strength: Effective in increasing CS and reducing neonatal mortality. Weaknesses: Poor implementation fidelity, lack of full cost coverage, inadequate communication, overburdened systems, and low HCW motivation.YesFocus on women of reproductive age in LMICs; evaluate government-led CS fee exemption policy with quantitative outcomes (CS rate, mortality)23.Witter et al. [23]SudanLower middle incomeFree Curative Care for Caesareans and Under-FivesUser fee exemption for priority maternal and child health servicesYesTo improve access and financial protection for curative care among pregnant women (Caesareans) and under-fivesMixed-methods (KIIs, exit interviews, facility survey, costing)Utilization rates, financial burden, quality of care, equity of access, health system impactsCS increased by 93% from 2006 to 2009; the average cost of CS was $135.6; significant inequities remained; policy suffered from poor implementation and fundingNo baseline or counterfactual data, fragmented HMIS, variable implementation, reliance on retrospective data, inconsistent financial recordsUtilization of CS rose post-policy; average CS cost ($135.6) remained unaffordable to 66% of women; stockouts, fragmented funding, inconsistent quality, weak monitoring, and facility-level inequities were evidentStrength: Increase in CS and service use. Weaknesses: Poor funding, unclear scope, exclusion of normal delivery, fragmented oversight, inequitable distribution, poor awareness, affordability issues despite the free policyYesFocus on government-led policy targeting CS in LMIC with outcome data24.Odunvbun et al. [32]NigeriaLower middle incomeFree Maternity Service PolicyRemoval of user fees for maternity servicesYesImprove maternal healthcare utilization and increase CS acceptance by removing cost barriersCross-sectional descriptive studyCS acceptability, previous CS history, and reasons for CS objectionCS acceptance was 60.6% (rural) and 68.3% (urban); 21% had previous CS; cost removal contributed to higher acceptanceCross-sectional design limits causal inference; only two facilities evaluated; self-reported attitudesQuant: CS acceptance averaged 64.5%; 21% had previous CS. Qual: Barriers included fear of pain, cultural beliefs, and perception of CS as failure.Strength: improved CS acceptance post-policy. Weakness: persisting myths, cultural beliefs, and fear of pain hinder the full impact.YesLMIC, government-led policy, CS outcome measured, original data, within timeframe25.Witter et al. [39]GhanaLower middle incomeNational Delivery Exemption PolicyUser fee exemption for delivery care in public and private facilitiesYesRemove financial barriers to the delivery of care and improve skilled birth attendance, including access to CSMixed-methods evaluation (household survey, utilization, clinical audit, funding analysis)Utilization, equity, CS cost coverage, OOP payments, quality of careUtilization increased, equity improved; CS costs reduced by 28%; the richest benefited more; funding gaps noted; quality of care remained a concernPartial cost recovery at the household level, underfunding of the scheme, inconsistent implementation, and limited quality improvementsQuant: CS costs fell 28%, deliveries rose; poorest quintile showed biggest gains. Qual: staff overworked, clients still incurred costs, care quality unevenStrengths: improved access, cost-effectiveness, equity. Weaknesses: underfunding, weak accountability, quality issues, the poorest not fully reachedYesLMIC, government-led, includes CS outcome and cost data, robust evaluation methodology, English full text, post-200026.Ajayi et al. [29]NigeriaLower middle incomeFree Maternal Healthcare PolicyUser fee exemption for maternal health, including CSYesImprove access to maternal services and reduce financial barriers, including CS.Cross-sectional population-based survey with logistic regression analysisCS prevalence and inequality by income, education, and residenceCS rate was 6.1%; significantly lower among poor, rural, and less educated women despite fee exemption. Higher odds of CS among women with income > N20,000 and higher education.Did not collect data on elective vs. emergency CS; health insurance status not included; did not assess provider-level decision makingQuant: CS rate 6.1%; richer, educated, urban women are more likely to access CS. Qual: persistent sociocultural, geographic, and health system barriers despite fee removal.Strength: addressed the cost barrier. Weaknesses: geographic inaccessibility, persistent inequalities, low provider capacity, cultural resistance, poor referral systemsYesLMIC, government-led, CS-related outcome, original data, recent, evaluates inequality post-policy27.Witter et al. Benin, Burkina Faso, Mali, MoroccoLMICs (all)Free or subsidized obstetric care, including CSsNational maternal health fee exemption/subsidy policyYesReduce maternal mortality by increasing access to facility-based deliveries and emergency obstetric care, including CSMixed-methods; health worker survey and facility dataHealth worker workload, satisfaction, motivation, availability, CS deliveries, policy awarenessIncreased CS rates, especially among poor women; improved satisfaction in 2 of 4 countries; increased workloads; limited staff training on policySelf-reported data, lack of baseline/control, underreporting of private practice, limited sample disaggregationCS increased (up to 200% in some cadres); midwives overburdened; access improved; perception of better drug/supply availability; lack of policy training; staff satisfaction variedStrengths: increased CS access, improved equity, and care quality. Weaknesses: inadequate HRH planning, lack of staff engagement/training, delayed reimbursementsYesGovernment-led policy in LMICs targeting CS access, with outcome evaluation and CS rate change data28.Dossou et al. [25]BeninLow incomeUser Fee Exemption Policy for Caesarean SectionMaternal health financing reformYesTo eliminate financial barriers to Caesarean section access by making CS free in public and some private hospitalsRealist evaluation (mixed-methods case study)Financial reimbursement rate per CS, kit completeness index, CS rates, trust levels, delays in reimbursementInitial success with high policy adherence (2009–2015), followed by declining support, reimbursement delays, and reduced implementation fidelity (2016–2018)Retrospective recall bias, limited generalizability, inadequate policy documentation, difficulty capturing full causal chainQuantitatively, CS kit completeness dropped from 95% to 17–38% after 2016; CS volume fell; funding ratio declined. Qualitatively, shifting political will, trust breakdown, and hospital autonomy challenges were key factors.Strengths: Rapid policy uptake, community trust, initial political backing. Weaknesses: Poor sustainability, lack of monitoring, loss of trust, delayed reimbursements, weakened enforcement, and fragmented hospital autonomy.YesGovernment-led, focused on CS, LMIC, measured outcomes, and implementation evaluation29.Galadanci et al. [12]NigeriaLower middle incomeFree Maternity Care Program in Kano StateUser fee removal for maternal servicesYesTo reduce maternal mortality by providing free antenatal, delivery, and Caesarean section services in public hospitalsDescriptive review (retrospective observational analysis)ANC attendance, hospital deliveries, CS rates, human resources, and infrastructure constraintsANC visits and CS rates increased significantly post-policy; the CS rate rose from 2.82% (2000) to 8.12% (2005). Utilization improved, but quality and access challenges persisted.No written policy; inadequate HRH, infrastructure, and funding; limited rural coverage; poor supply chainsCS rate increased from 2.82% to 8.12%. ANC visits more than doubled. Key constraints were workforce shortages, poor remuneration, inadequate infrastructure, and system-level supply issues.Strengths: Political continuity, state funding increase, access gains. Weaknesses: Human resource shortages, poor supply chains, limited PHC integration, and unaddressed demand-side barriers like low literacy.YesLMIC, government-led CS policy, original data on CS outcomes, retrospective evaluation of CS utilization trends30.Witter et al. [45]Benin, Burkina Faso, Mali, MoroccoBenin (LMIC), Burkina Faso (LIC), Mali (LIC), Morocco (UMIC)Free or subsidized delivery and Caesarean care policiesFee exemption/subsidy policy for obstetric careYesTo improve financial access to Caesarean and facility-based deliveries, and reduce maternal mortalityMixed-methods, realist case studyCaesarean rate, skilled birth attendance, household expenditure, financial protection, quality of careSignificant reduction in household payments; unclear impact on CS trends due to pre-existing positive trends; persistent quality and equity issues.No causal attribution is possible due to a lack of baseline/control; limited post-policy data pointsPolicies reduced CS-related household costs by up to 92%; CS rates increased, but not attributable solely to policy; continued gaps in neonatal care and facility quality.Strengths: political buy-in, some financial protection achieved. Weaknesses: unclear eligibility communication, poor quality of care, insufficient provider payment calibration, and partial implementation in some facilities.YesFocused on LMICs, government-led, assessed CS-related outcomes through mixed methods31.Edoka et al. [26]Sierra LeoneLowFree Health Care Initiative (FHCI)User fee removal and supply-side reforms (staffing, drugs, payroll)YesTo increase access and reduce out-of-pocket costs for maternal and child healthcare, including skilled birth attendance and facility deliveriesMixed-methods using Regression Discontinuity Design (RDD) and before-after time-trend adjusted designFacility delivery, delivery with skilled birth attendant, ANC visits, postnatal care, DPT3 vaccination, out-of-pocket costs5.4% increase in public facility delivery; 5.1–6% increase in skilled birth attendance; limited impact on actual CS rates; improvements diminished over timeNo control group for maternal outcomes; RDD based on intention-to-treat; no direct measurement of CS; potential reporting biasStatistical increase in skilled deliveries and ANC visits; qualitative findings suggest persistent informal fees and medicine shortagesStrengths: Measurable short-term gains in maternal care utilization. Weaknesses: Supply-side bottlenecks, limited impact on actual CS rates, and weak system accountabilityYesStudy evaluates government-led policy with outcome data relevant to CS proxy indicators (skilled birth, facility delivery)32.Lange et al. [19]BeninLowCS Exemption PolicyUser fee exemption policy for CS in public hospitalsYesTo reduce financial barriers to CS and increase access to emergency obstetric careQualitative ethnographic study with interviews and participant observationWomen’s experiences, CS-related costs, informal payments, perceptions of quality of careDespite the policy, women still incurred costs through informal payments and bribes. CS is viewed as life-saving but feared. Perceptions of care quality varied by facility, influenced by staff motivation and hospital leadership.Non-generalizable due to qualitative design and limited number of hospitals; findings are context-dependentQualitative data showed fee reduction improved access to CS, but persistent informal payments and mistreatment undermined benefits; care quality and CS access were uneven across hospitalsWeaknesses: Informal costs, coercion, variable leadership, and lack of respectful care. Strengths: Reduced financial burden and improved provider ability to refer for CS. Implementation is highly context-sensitive.YesEvaluates a government-led CS exemption policy with relevant qualitative outcome data on access, cost, and quality33.FEMHealt [44]BeninLowFree CS PolicyUser fee exemption for CS procedures in public and some private hospitalsYesTo increase access to obstetric care and reduce maternal mortality through free CS provisionMixed-methods assessment (quantitative service data, qualitative facility reviews)CS rate, patient expenditure, access to obstetric care, quality of care, equity of accessCS rate rose from 3.7% (2009) to 6.4% (2012); policy improved access, but mostly for middle and upper-income women; persistent out-of-pocket expenses reportedInconsistent free service coverage, poor quality of newborn care, implementation tools missing, inequity in access, persistent informal paymentsQuantitative: modest increase in CS rates and improved access. Qualitative: delays in care, poor newborn quality, high omission rates, inequities, lack of provider involvementStrengths: Improved facility funding, regular reimbursement, broader hospital coverage. Weaknesses: persistent costs, slow response times, exclusion of the poorest, lack of monitoringYesGovernment-led policy with measurable impact on CS rate and maternal healthcare utilization34.El-Khoury et al. [43]MaliLowCS Fee Exemption PolicyRemoval of user fees for CS procedures in public sector facilitiesYesTo reduce financial barriers and improve access to life-saving obstetric care by providing free CSsCross-sectional patient survey with wealth index comparison against DHS dataCS access distribution by socioeconomic status, wealth quintile disparities, and overall utilizationCS utilization increased post-policy, but wealthier women (top 40%) received 58% of CS procedures. The poorest 40% received only 27%, despite making up 45% of deliveries.No baseline (pre-policy) comparison; CS and birth data collected 4 years apart; limited asset indicators; regional disparities not fully accounted forQuantitative: Disproportionate access to CS among wealthier women. Qualitative: Barriers like transport cost, incomplete CS kits, and low awareness among poor women limit the impact.Strengths: Policy increased national CS rates and reduced direct costs. Weaknesses: Inequitable access persists due to transport, supply, and informational barriers not addressed.YesGovernment-led CS fee exemption policy with outcome data on CS rate distribution and health equity impact35.Fournier et al. [11]MaliLow incomeFree Caesareans Policy (Free-CSec)Fee exemption policy for CSsYesReduce maternal and neonatal mortality by removing financial barriers to CSsInterrupted Time Series (ITS)CS rate by area of residence, AMI-based Caesarean proportionCS rates increased from 0.25% to 1.5% overall; 1.7% to 5.7% in cities with hospitals; little impact in rural areas.Geographic inequities, incomplete coverage of transportation/accommodation costs, some overuse risksQuantitative: Increase in CS rates in urban areas; limited or no change in rural areas. Qualitative: Financial and geographic access barriers persist.Strength: Improved access in urban hospitals. Weakness: Persistent rural inequities; partial financial coverage; decline in AMI-Caesarean proportion over time.YesGovernment-led policy aimed at influencing CS use with outcome evaluation in an LMIC36.Ganaba et al. [13]Burkina FasoLow incomeObstetric Care Subsidy PolicySubsidy for facility-based deliveries and EmOC (partial fee exemption)YesReduce maternal mortality, improve access to delivery services, and enhance the quality of obstetric careMixed-methods (quantitative + qualitative) with segmented regression and case studiesCS rate, facility-based deliveries, household costs, omission scores, implementation quality, health worker motivationFacility-based deliveries increased significantly post-policy, but CS rates rose slightly and not significantly; equity in CS access remains limitedLack of baseline data; absence of control group; varying local implementation; continued household costs; weak indigent targetingQuantitative: 4% annual increase in facility deliveries; small increase in CS use post-policy, but not statistically significant. Qualitative: policy awareness is low, inequities in access remain, and staff reported mixed motivation due to administrative burden and lack of financial incentives.Strength: Policy reduced delivery costs and increased service use. Weakness: Poor implementation fidelity, inequity in CS access, and non-compliant practices undermined impact, especially in regional hospitals.YesGovernment-led, LMIC-based, includes CS-related outcomes, evaluation of policy impact with mixed methods37.Nguyen et al. [17]BangladeshLower middleBangladesh Voucher Program for Maternal HealthDemand-side financing; voucher-based subsidyYesTo increase maternal health service utilization and reduce financial barriers to institutional delivery and skilled attendanceQuasi-experimental (cross-sectional, difference-in-differences, mother-fixed effects)ANC visits, institutional delivery, skilled birth attendance, OOP cost, Caesarean sectionSignificant improvements in use of ANC, skilled birth attendance, and institutional delivery; 64% reduction in OOP cost; no significant impact on Caesarean section rateNon-randomized design, recall bias in past birth data, limited quality data, and potential selection biasQuantitative: Voucher program led to +46% use of skilled providers and +14% institutional delivery; no effect on CS rate. Qualitative: Vouchers are well-accepted, improved awareness, but challenges with reimbursement and quality of careStrengths: Highly targeted to poor women; mixed demand/supply incentives. Weaknesses: Limited effect on CS, delayed reimbursements, supply-side capacity constraintsYesGovernment-led, focused on women of reproductive age in LMIC, includes CS-related outcomes, and has a quantitative evaluation design


These patterns highlight a literature that is both policy-rich and methodologically diverse, yet still geographically skewed, with little representation outside Africa. This limitation is important for interpreting the generalizability of subsequent findings and is further discussed later in the manuscript.

### 3.2. Service Utilization and Coverage Effects

Twenty-two studies (59%) documented a significant rise in the volume of CS following policy implementation. In Senegal, the national free exemption policy aimed at removing financial barriers to facility-based childbirth and CS was found to increase the CS rate from 4.2% to 5.6% within three years [40]. Another complementary qualitative assessment of slum residents in Dakar, Senegal, found persistently high out-of-pocket (OOP) payments, as many poor women still pay significant costs despite the policy. Costs vary widely by location and facility, and delayed state reimbursements were cited as a cause [37]. In Mali and Benin, another study reported a 36% pooled relative increase in CS utilization after the abolition of user fees [20], and similar findings were reported by another time series analysis from Kayes region, where CS rates rose from 0.25% to 1.5% nationally and 5.7% in hospital catchment cities [11]. In another study from Kano state, Nigeria, the subnational government rolled out a policy to reduce maternal mortality by providing free antenatal, delivery, and CS services in public hospitals, which was reported to have tripled the utilization of CS from 2.8% to 8.1% over five years [12]. Moreover, in another subnational study from Delta State, Nigeria, removal of user fees for maternity services, including CS, led to a self-reported CS acceptance rate of 64.5% which was higher in the urban compared to rural areas [32].

### 3.3. Service Utilization and Coverage

Across the evidence base, 22 of the 37 studies (59%) documented significant increases in CS utilization following the launch of government-led policies. In Senegal, for example, the Free Delivery and Caesarean Policy increased CS rates from 4.2% to 5.6% within three years [41], yet a complementary study in Dakar’s urban slums revealed that many women continued to pay an average of US$296 because of delayed government reimbursements [37]. Similar positive patterns were evident in Mali and Benin, where pooled analyses reported a 36% rise in CS utilization [27]. Another interrupted time series from Kayes showed national rates increased from 0.25% to 1.5% overall and from 1.7% to 5.7% in urban hospital catchments [11]. In Nigeria, Kano State’s free maternity policy tripled the utilization of CS from 2.8% to 8.1% over five years [12], while a study in Delta State found that 64.5% of women reported acceptance of CS, with uptake higher in urban than rural areas [32].

Sector-wide reforms, defined as multi-component, system-level packages that go beyond fee exemptions by combining changes to financing/provider payment, benefit packages, human resources, infrastructure & supplies/medicines, service delivery organization, and governance/data systems, implemented together as a health-system reform bundle with CS/maternal services embedded, delivered mixed effects. The Tanzania’s exemption policy raised CS from 2.5% to 4.8% but widened rural–urban disparities [28], while in Kenya, the 2013 Free Maternity Policy raised CS by 28.9% in the public sector before gains eroded under the subsequent Linda Mama insurance scheme [30,33].

Outside Africa, results were similarly varied. In Iran, the Health Sector Evolution Plan produced a temporary drop in CS rates followed by a rebound [14], while a complementary reform in Shiraz reduced CS prevalence from 64.7% to 58.6%, though the change was not statistically significant [22]. In Georgia, the introduction of punitive financial penalties reduced national CS rates from 44.7% to 40.8% without adverse effects on perinatal outcomes [36]. Demand-side financing schemes were less consistent. Bangladesh’s national voucher program increased antenatal care and institutional deliveries but had no measurable effect on CS utilization [16,17]. Similarly, Burkina Faso’s subsidy schemes significantly raised facility-based births but had only modest, and in many cases non-significant, impacts on CS rates, while more than half of women undergoing CS continued to pay at the point of care [31,45].

### 3.4. Equity, Financial Protection, and Hidden Costs

Equity gains were limited. About 14 of the 26 fee-exemption studies (54%) found evidence of pro-rich or pro-urban drift. In Mali, the wealthiest 40% of women received 58% of CSs while the poorest 40% received only 27% despite representing nearly half of deliveries [43]. Similar inequities were observed in Benin [25], Nigeria [29], and Sudan [23]. By contrast, a quasi-experimental evaluation from Mali and Benin suggested larger absolute gains among rural and non-educated women compared with the wealthiest groups, particularly in Mali [27].

Financial protection impacts remained elusive. Despite nominally free policies, catastrophic expenditure persisted. In Mali, between 40% and 43% of households undergoing CS experienced catastrophic spending largely driven by drug purchases and transport costs [7,47]. In Burkina Faso, median out-of-pocket payments for CS remained US$13.78 [31], while in Senegal’s slums women reported paying nearly US$296 [37]. Morocco’s policy reduced official CS charges by 40% in SEGMA hospitals but still left households with average OOP costs of US$169 [42]. Informal or under-the-table payments were commonly reported in Benin [19,44], Burkina Faso [21], and Ghana [39], often linked to reimbursement delays [25,45] and persistent supply shortages [47].

### 3.5. Health Outcomes and Quality

Mortality outcomes were less frequently studied. A difference-in-differences analysis in Mali and Benin found a 30% reduction in neonatal mortality associated with the removal of user fees [46]. In contrast, Georgia’s punitive policy showed no measurable impact on perinatal mortality [36], while Kenya’s Free Maternity Policy generated sizeable increases in the utilization of emergency obstetric care but no change in CS or stillbirth rates [30]. No study directly assessed impact of CS policy on maternal mortality.

Quality-of-care findings were mixed. Over-medicalization was found to be a major concern in several contexts. In Benin and Mali, CS was sometimes driven by maternal distress, lack of pain relief, or provider incentives rather than medical indications [34]. In China, the transition from a one-child to a two-child policy coincided with a high prevalence of repeat CS, particularly among older mothers, with rates reaching 64.8% in women aged 40 years or more [18]. Underuse also persisted, especially where systemic bottlenecks limited surgical capacity. In Bangladesh, for example, of the 43 women eligible for CS under the voucher scheme, only 10 underwent the procedure locally while the rest were referred elsewhere because of staff shortages and poorly equipped theatres [16]. Similar supply-side constraints were evident in Nigeria [13] and Burkina Faso [23], where health worker surveys highlighted rising workloads, low morale, and insufficient training following the roll-out of maternal health financing policies.

### 3.6. Implementation Dynamics

Implementation of fidelity was found to be a decisive determinant of outcomes. Delayed reimbursements were the most consistent obstacle, with evidence from Senegal [41], Benin [25], Burkina Faso [21], Kenya [35], Mali [47], and Sudan [23] all documenting reimbursement delays that forced facilities either to ration supplies or pass costs on to patients. In Sudan, reimbursements per CS fell below breakeven thresholds and as a result, most health facilities were unable to sustain CS and other services, and compelled patients to pay despite the formal fee exemption policy [23].

Communication failures were also widespread. In Burkina Faso, only 37% of women were aware of which drugs were covered under the free care policy [31]. In Kenya, facility managers expressed uncertainty about whether the Linda Mama scheme guaranteed reimbursement for CS [33], while in Benin there was no formal codification and providers continue to collect discretionary charges from patients [24].

Supply-side readiness consistently shaped impact trajectories. Policies that paired user-fee exemptions with parallel investment in infrastructure and human resources, such as Tanzania’s Health Sector Reform [28] and Iran’s Health Sector Evolution Plan [14], achieved stronger initial improvements in CS utilization. However, in contexts where exemptions were introduced without adequate supply-side investment, the early gains proved unsustainable. For example, in Benin, reimbursement delays lengthened from 26 to 259 days, kit completeness declined from 95% to as low as 17–38%, and CS volumes subsequently declined [25].

## 4. Discussion

This review shows that the removal of user fees during childbirth remains the most widespread government strategy to address low CS coverage in LMICs. Two consistent patterns were noted from the evidence. First, abolishing direct charges for surgery can produce an immediate and visible surge in CS uptake. Second, the benefits of these reforms are unevenly distributed, often bypassing the poorest women while generating new challenges for financial protection, quality, and sustainability.

Early spikes in utilization are striking. Policies in Senegal, Mali, Benin, and Kano State in Nigeria all reported absolute or relative gains that doubled or even tripled baseline CS volumes within a few years of implementation [11,12,28,46]. Tanzania’s sector-wide reforms produced similar effects, while Kenya’s Free Maternity Policy drove rapid increases that eroded under the budgetary strains of Linda Mama [28,30,33]. These findings validate economic theory that reducing the price of care will shift the demand outward, but they also highlight a critical weakness in which higher utilization is not synonymous with equitable or sustained access.

The results of equity outcomes remain less convincing. Many evaluations documented a pro-urban or pro-rich drift, as seen in Mali where the richest 40% of women accounted for 58% of procedures compared with 27% among the poorest [43]. Similar gradients were found in Benin, Sudan, and Nigeria [23,25,29]. Ghana’s long-running fee exemption program demonstrated that skilled birth attendance increased, but gains were concentrated among wealthier quintiles [50]. Only a quasi-experimental study spanning Mali and Benin provided counter-evidence and showed greater improvements among rural and uneducated women [27]. These results illustrate that fee abolition alone cannot solve the problems associated with hospital distance, transport, health literacy, and entrenched gender norms. Experiences from Burkina Faso reinforce this message. Despite national subsidies, CS rates remained the same in districts with inadequate surgical staff [13]. Complementary interventions such as maternity waiting homes, transport vouchers, and targeted communication campaigns are, therefore, not optional but necessary elements of equity-oriented CS policy design.

Financial protection was another area where expectations were not met. Catastrophic health spending persisted in Mali, Burkina Faso, and Sierra Leone because families continued to pay for medicines, blood transfusion, consumables, or transportation [7,31,51]. Collection of informal fees from patients became widespread in settings where government reimbursements were delayed, a recurrent problem reported in Senegal, Benin, and Sudan [23,25,45]. Evidence from India’s Ayushman Bharat scheme illustrates the same lesson in a different context. Despite the wide insurance coverage, CS costs continued to rise because of fraudulent billing and double charging practices [52]. These findings highlight that achieving true protection requires not only removing fees at the point of care, but also enforcing timely reimbursements, transparent procurement, and robust oversight of provider behavior.

The tension between underuse and overuse is also evident. Evidence for unnecessary CS section is mainly qualitative (provider incentives, maternal distress, weak supervision) and only few studies directly measure clinical appropriateness. In Benin and Mali, hospitals rapidly transitioned from severe undersupply to CS rates exceeding 40%, often driven more by provider preference than clinical necessity [34]. In urban China, CS among women aged forty or older reached nearly 65% following the two-child policy, largely attributable to repeat procedures [18]. These patterns suggest that when patients face no direct cost, but providers benefit from fee-for-service incentives, supplier-induced demand becomes inevitable. By contrast, Georgia’s punitive tariff system reduced CS rates from 44.7% to 40.8% without a negative impact on perinatal outcomes [36]. Comparative evidence from bundled and capped payment models shows that carefully designed disincentives can restrain unnecessary surgery, but require careful monitoring to avoid harmful under-provision [15].

The success or failure of financial reforms was closely tied to service readiness. Tanzania combined its CS reforms with investments in theatres and anesthesia, and this led to a tripling of national CS rates [28]. Iran’s Health Transformation Plan initially reduced unnecessary surgery by promoting vaginal deliveries and improving the quality of maternity services, but rates later rebounded when women perceived declining standards of intrapartum care [14,53]. In Bangladesh, where facility readiness was inadequate, voucher recipients often required referral elsewhere, undermining both access and trust [16]. In Benin, reimbursement delays undermined supply chains and the share of required items available in standard CS kits dropped from 95% to less than 40%, followed by a decline in CS volumes [25]. These examples demonstrate that financial reforms cannot succeed in isolation, and they must be accompanied by parallel investments in infrastructure, workforce, and supply chains.

Delayed reimbursement was found to be the most consistent impediment to effective policy. Delays or shortfalls in reimbursements led to the re-emergence of informal payments, a decline in staff morale, and stagnation in service utilization. Poor communication compounded these effects, leaving clinicians uncertain about which procedures qualified, and women unsure of what was genuinely free [31,35]. In contrast, Nepal’s program, which combined transparent reimbursement with clear entitlements, achieved a fivefold increase in institutional deliveries, though benefits accrued disproportionately to wealthier districts [52]. These cases align with organizational change theory, which asserts that policy effectiveness depends not only on design, but also on the operational culture and incentives shaping implementation [10].

### Limitations

Several structural and methodological limitations constrain interpretation. Only six studies employed high-frequency interrupted time series capable of distinguishing policy effects from secular trends. Mortality outcomes were rarely reported, leaving uncertainty about the ultimate impact on maternal survival. Private sector dynamics, especially in South Asia, were poorly captured despite their dominance in obstetric care [38]. Few studies explicitly examined context–mechanism–outcome pathways using realist or mixed-methods designs, with the exception of research from Benin [24]. The language restriction to English also introduces bias. The lack of GRADE or comparable approaches limits the strength of policy recommendations. This review is a narrative synthesis and did not transform data to enable pooled effect-size estimates, limiting the ability to quantify strategy effectiveness. In addition, the evidence base is concentrated in African countries, which constrains generalizability to other LMIC regions where obstetric care systems differ. These gaps underscore the need for future research to link fiscal flows to clinical outcomes using counterfactual designs and should employ broader search strategies, multilingual inclusion, and structured certainty grading.

## 5. Conclusions

Our findings suggest five lessons for policymakers. First, eliminating user fees is a necessary but insufficient measure unless paired with reliable referral, supply, and blood transfusion systems. Second, reimbursement must be timely and match actual costs to avoid the resurgence of informal charges. Third, aligning provider incentives with clinical appropriateness through bundled payments or audit-feedback mechanisms is essential to balance access with restraint. Fourth, targeted equity instruments such as transport vouchers and transparent eligibility criteria are indispensable for reaching the poorest. Finally, real-time monitoring that tracks utilization, equity, quality, and expenditure enables mid-course corrections and strengthens accountability. In a cross-case comparative synthesis, we distilled a practical design–governance framework that links financing instruments to utilization, equity/financial protection, and quality/safety outcomes (Figure 4).

Policy design levers (financing instrument, provider payment/purchasing, equity instruments) require implementation enablers (service readiness, governance/operations, financing execution) to activate three mechanisms of action, namely price signal, supply readiness & incentives, and accountability/timely reimbursement leading to utilization, equity, financial-protection, and quality/safety outcomes. Safeguards (e.g., Robson monitoring, indication audits, redressal of grievances, equity dashboards) and feedback loops (dashboards, rapid evaluations, adaptive tariff/revision of CS kit) enable data-informed course correction. Across settings, fee-removal plus (i) theatre/anesthesia investment, (ii) on-time reimbursement matched to true cost, and (iii) explicit equity instruments (transport vouchers, indigent criteria) yielded larger, more equitable gains (e.g., Tanzania, parts of Mali/Benin). Fee-removal alone often produced urban/pro-rich drift and hidden costs (e.g., Burkina Faso, Sudan, Senegal) while payment disincentives can reduce overuse (Georgia) when coupled with monitoring. While funding reforms for CS are critical, the systems through which funds flow ultimately shape outcomes. Countries that integrated financial relief with infrastructure, workforce readiness, and equity safeguards made tangible progress toward safe access to CS. Those that focused narrowly on fee removal often replaced one barrier with another, as women are confronted with barriers of persistent distance, informal payments, or overuse. The evidence reinforces that sustainable improvements in maternal health depend not on abolishing user fees alone, but on embedding financing reforms within a broader architecture of readiness, accountability, and equity.

## Figures and Tables

**Figure 1 healthcare-13-02522-f001:**
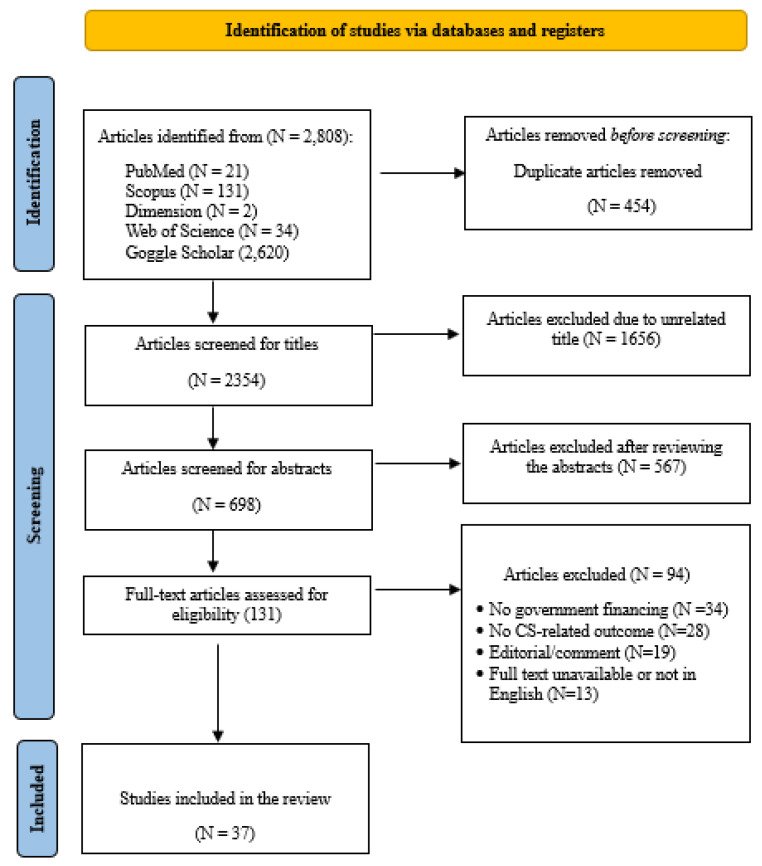
PRISMA flow diagram of the study selected.

**Figure 2 healthcare-13-02522-f002:**
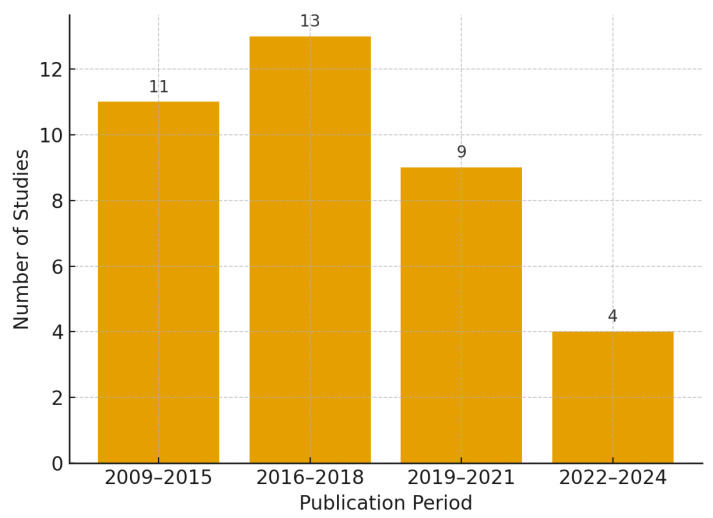
Number of Studies by Year of Publication.

**Figure 3 healthcare-13-02522-f003:**
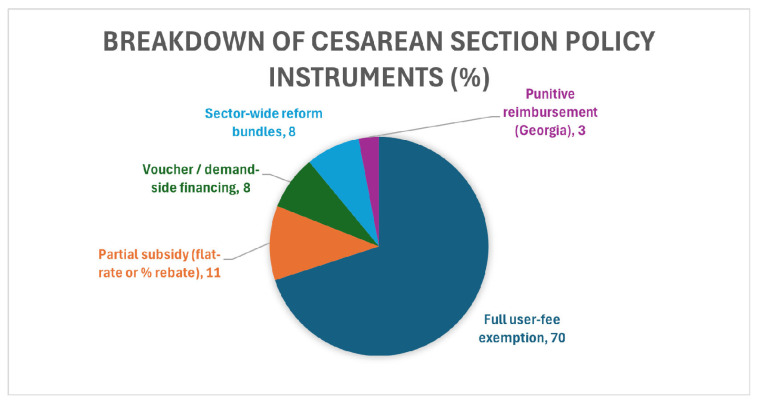
Breakdown of Caesarean Section Policy Instruments across LMIC (%).

**Figure 4 healthcare-13-02522-f004:**
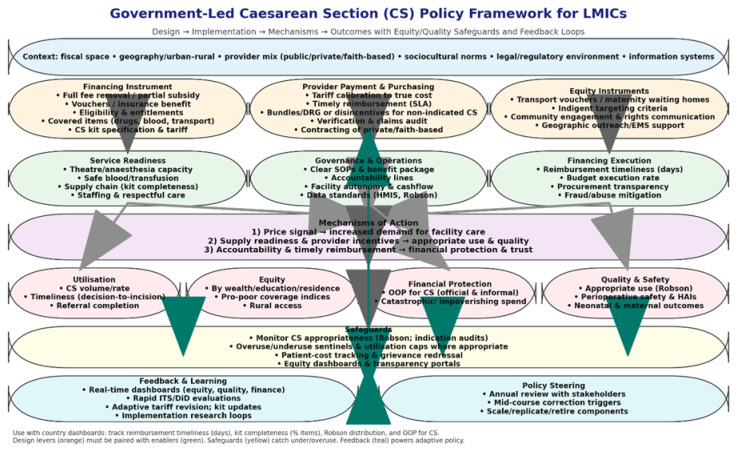
Government-led Caesarean section (CS) policy design & implementation framework for LMICs.

**Table 1 healthcare-13-02522-t001:** Eligibility framework (Inclusion and Exclusion Criteria).

S/N	Criterion	Included Studies	Excluded Studies
1.	Population	Women of reproductive age in World Bank-classified LMICs during the study period	Populations in high-income settings
2.	Intervention /exposure	Government-led or government-initiated policies that directly or indirectly altered user charges for CS or delivery care, fee exemption, partial subsidy, national health insurance, vouchers, provider-payment reform, or broader sector reforms that embedded CS financing	Purely private sector initiatives, clinical trials without a policy component, or supply-only quality-improvement projects
3.	Comparison	Any design was accepted, including pre-policy baselines, contemporaneous controls, or interrupted time series without a formal control.	Opinion pieces, editorials, or studies with no outcome evaluation
4.	Outcomes	CS utilization, facility delivery, maternal or neonatal health outcomes, equity gradients, out-of-pocket (OOP) spending, implementation processes	Articles that mentioned CS only incidentally or reported no CS-related outcome
5.	Study type	Quantitative, qualitative, or mixed-methods primary research published in English, 1 January 2009–30 May 2025	Reviews without original data, non-English texts, conference abstracts without full papers

## Data Availability

The original contributions presented in this study are included in the article/Appendix A. Further inquiries can be directed to the corresponding author.

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
