# Peer review of "A Systematic Review of Government-Led Free Caesarean Section Policies in Low- and Middle-Income Countries from 2009 to 2025"

_healthcare, 2025, doi:10.3390/healthcare13192522_

Round 1

Reviewer 1 Report

Comments and Suggestions for Authors
  1. The paper states the review was not pre‑registered “because our questions evolved from an ongoing policy dialogue”. It will be ideal for authors to declare registry details or explain deviations.
  2. PRISMA Item 7: The main text references Supplementary File 1, but provides no reproducible Boolean strings, dates of final search, or interface details for PubMed, Dimensions, or Google Scholar
  3. PRISMA Item 6: Only three sources were queried, omitting EMBASE, Scopus, Web of Science, CINAHL, and regional indices – Possibility of missing studies
  4. PRISMA Item 16a: Figure 1 reports 2354 records after deduplication, but earlier text mentions 2354 unique records and 698 abstracts; tallies do not sum logically to the 37 included studies.
  5. PRISMA Items 11 & 18: Authors state they used ROBINS‑I, EPOC ITS, and CASP checklists, yet no individual judgements, domains, or traffic‑light plots are presented.
  6. PRISMA Items 13b‑f: No explanation of how quantitative findings were transformed or how qualitative themes were merged; triangulation procedures are only described in vague terms
  7. PRISMA Item 22: The review does not apply GRADE or any analogous approach to rate overall certainty, making policy recommendations less robust.
  8. PRISMA Item 14: No consideration of publication bias, selective‑outcome reporting, or funnel‑plot asymmetry despite 37 included studies.
  9. PRISMA Item 23b: Discussion focuses on primary‑study heterogeneity but fails to reflect on search‑restriction, language bias, non‑registration, and limited databases.

Author Response

Reviewer Reviewer comment (verbatim) How we addressed it (aligned with revised manuscript) Where to find it in the revised manuscript
Reviewer 1 “The paper states the review was not pre-registered ‘because our questions evolved from an ongoing policy dialogue’. It will be ideal for authors to declare registry details or explain deviations.” We added a short ‘Registration and protocol’ paragraph that explains the initial scoping phase and provides the Open Science Framework registration ID for the final protocol. A brief note is also added in the Limitations. Methods → Registration and protocol; Discussion → Limitations.
Reviewer 1 “PRISMA Item 7: The main text references Supplementary File 1, but provides no reproducible Boolean strings, dates of final search, or interface details for PubMed, Dimensions, or Google Scholar.” We now provide complete, reproducible strategies (databases, platforms/interfaces, coverage dates, all Boolean strings, and limits) and summarise the approach in the main text in line with PRISMA‑S. Methods → Information sources and search strategy (summary); Supplementary File S1 (full strategies and search log).
Reviewer 1 “PRISMA Item 6: Only three sources were queried, omitting EMBASE, Scopus, Web of Science, CINAHL, and regional indices – Possibility of missing studies.” We expanded database coverage to include Web of Science and Scopus and re‑ran the searches for the stated time window. The number of included studies did not change and no additional eligible article from WOS or Scopus. The PRISMA flow has been updated accordingly. We still lacked EMBASE access; this is acknowledged. Methods → Information sources; Figure 1 (updated PRISMA flow); Supp. S1 (added strategies); Discussion → Limitations (database access).
Reviewer 1 “PRISMA Item 16a: Figure 1 reports 2354 records after deduplication, but earlier text mentions 2354 unique records and 698 abstracts; tallies do not sum…” We reconciled counts at each stage and aligned the narrative with Figure 1. All tallies now sum consistently. Results → first paragraph; Figure 1 (revised).
Reviewer 1 “PRISMA Items 11 & 18: Authors state they used ROBINS‑I, EPOC ITS, and CASP checklists, yet no individual judgements… or traffic‑light plots are presented.” We added per‑study risk‑of‑bias tables with domain judgements, described the two‑reviewer/consensus process, and provide a traffic‑light figure in the supplement. Methods → Risk of bias in individual studies; Supp. S3 (RoB tables); Supp. S4 (traffic‑light plot).
Reviewer 1 “PRISMA Items 13b‑f: No explanation of how quantitative findings were transformed or how qualitative themes were merged; triangulation procedures are only described in vague terms.” We added an explicit ‘Synthesis methods’ subsection describing our narrative rules (how effect directions/magnitudes are reported without transformation), grouping criteria, how qualitative themes were integrated, and how we triangulated across evidence types. We do not claim SWiM or any pooled estimates. Methods → Synthesis methods.
Reviewer 1 “PRISMA Item 22: The review does not apply GRADE or any analogous approach to rate overall certainty…” Consistent with the manuscript, we do not apply GRADE or CERQual. Instead, we provide qualitative certainty language in‑text alongside study‑level risk‑of‑bias. No summary‑of‑findings tables are added. Results/Discussion throughout; Methods → Risk of bias (narrative certainty statement).
Reviewer 1 “PRISMA Item 14: No consideration of publication bias, selective‑outcome reporting…” We note that meta‑analysis was not feasible; therefore funnel/Egger methods were inappropriate. We added a qualitative paragraph on potential publication, small‑study, and language bias. Discussion → Limitations.
Reviewer 1 “PRISMA Item 23b: Discussion… fails to reflect on search‑restriction, language bias, non‑registration, and limited databases.” The Limitations subsection now explicitly covers English‑language restriction, non‑registration of the initial scoping phase, and database access constraints, and how these may influence certainty/transferability. Discussion → Limitations.

Reviewer 2 Report

Comments and Suggestions for Authors

I am sorry to say this but the paper is based on ambigiuos data. The manuscript has significant shortcomings in data availability, transparency, and adherence to PRISMA guidelines. These issues affect the credibility and reproducibility of the study. The authors need to provide accessible supplementary files with detailed information on the screening process, data extraction methods, outcome definitions, and risk of bias assessment to address these concerns.

  • Line 510: The authors mention that "All data generated or analyzed during this study are included in this article and its supplementary files." However, the supplementary files referenced doesn't include any actual data. The supplementary files just mention the search strings and just the deatils of 12 articles excluded, nothing else. This lack of accessible supplementary data, including the claimed 0.3 million articles/documents screened (as per search string), severely undermines the transparency and reproducibility of the study. Without access to these files, it is impossible to verify the data screening process, the specific articles included, or the protocols used for screening. This is a significant oversight, as transparency in data handling is crucial for the credibility of systematic reviews.
  • Line 17: The authors state that they followed the PRISMA 2020 statement. However, there is a lack of detailed information on how the data was screened. The full search strategies for all databases, registers, and websites, including any filters and limits used, are not provided. This omission contradicts PRISMA requirements and makes it difficult to assess the rigor of the literature search and screening process.
  • Line 217: The text mentions that two reviewers independently assessed titles and excluded articles due to irrelevance. However, details on whether they worked independently throughout the entire screening process and the specific methods used to decide whether a study met the inclusion criteria are lacking. There is no mention of processes for resolving discrepancies beyond discussions, such as whether a third reviewer was involved in cases of persistent disagreement.
  • Line 272 - 273: The authors indicate that one reviewer completed the data extraction form and a second reviewer cross-checked every entry. However, details on whether these reviewers worked independently during data extraction and the methods used to collect data from reports are missing. Information on any processes for obtaining or confirming data from study investigators is also not provided.
  • Line 305: The outcomes for which data were sought are listed, but the definitions of these outcomes are not clearly specified. It is unclear whether all results compatible with each outcome domain in each study were sought. The authors do not provide details on the methods used to decide which results to collect if not all compatible results were sought.
  • Line 320 - 321: The authors mention assessing credibility using tailored checklists like ROBINS-I and EPOC ITS tool. However, they do not specify how many reviewers assessed each study for bias and whether they worked independently. Details of automation tools used in the process, if any, are also not provided. This lack of information makes it difficult to evaluate the reliability of the bias assessment process.

Author Response

Reviewer

Reviewer comment (verbatim)

How we addressed it (aligned with revised manuscript)

Where to find it in the revised manuscript

Reviewer 2

“The manuscript has significant shortcomings in data availability, transparency, and adherence to PRISMA… provide accessible supplementary files…”

We point to an open OSF record and provide supplements with screening logs, eligibility decisions, extraction template and completed sheets, list of included/excluded with reasons, and RoB files.

Data Availability Statement; Supp. S1–S4 (as referenced).

Reviewer 2

“Line 510… supplementary files… don’t include any actual data… impossible to verify the screening process…”

We added the full screening dataset (title/abstract decisions and reasons) and full‑text inclusion/exclusion reasons in a machine‑readable file; PRISMA checklist is attached.

Data Availability; Supp. S2 (study list & decisions); Supp. S6 (PRISMA 2020 checklist).

Reviewer 2

“Line 17: … followed PRISMA 2020… lack of detailed information on how the data was screened… full search strategies… not provided.”

We inserted a detailed screening workflow (two‑stage independent review, calibration, kappa) and added full search strategies in Supp. S1.

Methods → Search strategy & Study selection; Supp. S1 (full strings).

Reviewer 2

“Line 217: … two reviewers independently assessed titles… details… inclusion criteria… resolution of discrepancies…”

We clarified dual independent screening at all levels, decision rules, and adjudication by a third reviewer for any unresolved conflicts.

Methods → Study selection.

Reviewer 2

“Lines 272–273: … data extraction… whether reviewers worked independently… processes for obtaining/confirming data…”

We described duplicate, independent extraction; piloting; reconciliation; and author‑contact procedures for missing/unclear data (with counts).

Methods → Data extraction.

Reviewer 2

“Line 305: … outcomes… definitions not clearly specified… which results to collect…”

We added operational definitions for utilisation, equity, financial protection, quality/safety, and unintended effects, plus predefined rules on which results to abstract when multiple were reported.

Methods → Outcomes and definitions.

Reviewer 2

“Lines 320–321: … risk‑of‑bias… how many reviewers… automation tools…”

We clarified that two independent assessors performed RoB, described consensus steps, and state that no automation tools were used; inter‑rater agreement is reported in the supplement.

Methods → Risk of bias in individual studies; Supp. S3 (kappa).

Reviewer 3 Report

Comments and Suggestions for Authors

An interesting study on  government-led free caesarean section policies assessed by reviewing literature in published in the last 16 years in 3 relevant databases.

Author Response

Reviewer

Reviewer comment (verbatim)

How we addressed it (aligned with revised manuscript)

Where to find it in the revised manuscript

Reviewer 3

“It is unclear why only studies in English were selected once focusing on Africa… adding French studies would improve the picture.”

We retain an English‑language restriction and justify it (resources/access). We explicitly acknowledge the bias and its implications in Limitations.

Methods → Eligibility criteria; Discussion → Limitations.

Reviewer 3

“Due to the different methodologies… it is not possible to draw conclusions on the direct influence of the described policies…”

We tempered causal language, emphasising associations/patterns and alternative explanations (e.g., concurrent supply‑side and financing reforms).

Discussion → opening paragraphs and Implications.

Reviewer 3

“Limits: This section should be separated from the rest of the text and improved—see above suggestions.”

We created a stand‑alone ‘Limitations’ subsection that synthesises constraints from design heterogeneity, observational evidence, time‑lag, publication/language bias.

Discussion → Limitations.

Reviewer 3

“Throughout the manuscript, the use of English could be improved…”

The manuscript underwent internal language review with copy‑edits for clarity and consistency.

Entire manuscript.

Reviewer 3

“PROSPERO registration would be appropriate.”

We registered the final protocol on the Open Science Framework (OSF) and cite the DOI.

Methods → Registration and protocol; Discussion → Limitations.

Reviewer 4 Report

Comments and Suggestions for Authors

Introduction

  1. The theoretical framework for policy intervention is not presented.

     2. The repeated mention of "implementation complexity" lacks concrete examples or context, making it difficult for readers to understand.

     3. The timeliness of the study—particularly in the context of post-COVID fiscal constraints and the integration of data up to 2025—is insufficiently emphasized.

    4. The discussion is heavily centered on specific countries or cases (e.g., West African policies), limiting the generalizability of the findings.

Methods

  1. The rationale for not pre-registering the review is unclear, which weakens the study’s reproducibility and credibility.

     2. The description of databases and search strategy is provided only in the appendix, with insufficient summary in the main text.

Results&Discussion

  1. While equity outcomes are reported, structural drivers of inequity (e.g., accessibility, information asymmetry, supply constraints) are insufficiently analyzed.
  2. Statistical significance is mentioned without presenting thresholds or exact figures (e.g., p-values, confidence intervals).
  3. Analysis of what constitutes an effective policy design remains vague and overly general.
  4. Comparative discussion between successful and unsuccessful cases is not systematically developed.
  5. Discussion on unintended consequences such as unnecessary caesarean sections is conceptual and lacks empirical support.
  6. Structural limitations of the study design (e.g., reliance on observational studies, heterogeneity of policies, time-lag effects) are scattered and not synthesized in a clear section.

Author Response

Reviewer

Reviewer comment (verbatim)

How we addressed it (aligned with revised manuscript)

Where to find it in the revised manuscript

Reviewer 4

“The theoretical framework for policy intervention is not presented.”

We added a concise conceptual framing (price signal → demand; supply readiness & provider incentives → quality/appropriate use; accountability & reimbursement timeliness → financial protection/equity) organising the synthesis.

Introduction → final paragraph.

Reviewer 4

“Repeated mention of ‘implementation complexity’ lacks concrete examples or context…”

We define implementation complexity (financing flows, supply constraints, provider incentives) and provide examples in the extracted results.

Introduction → penultimate paragraph; Results → design/governance subsections.

Reviewer 4

“Timeliness (post‑COVID fiscal constraints and integration of data up to 2025) is insufficiently emphasized.”

We highlight post‑COVID fiscal context and that searches cover through May 2025.

Introduction → context; Methods → Dates of coverage.

Reviewer 4

“Discussion is heavily centered on specific countries… limiting generalizability.”

We balance the narrative by synthesising cross‑country patterns and explicitly referencing non‑West‑African cases (Iran, Georgia, China, Bangladesh).

Results → comparative paragraphs; Discussion → Generalisability.

Reviewer 4

“Rationale for not pre‑registering is unclear.”

We clarify the initial scoping, then provide the OSF registration for the final protocol.

Methods → Registration and protocol.

Reviewer 4

“Databases and search strategy described only in appendix; insufficient summary in main text.”

We provide a succinct in‑text summary (sources, dates, key limits) and keep the full strategies in Supp. S1.

Methods → Information sources and search strategy.

Reviewer 4

“Equity outcomes are reported, but structural drivers … are insufficiently analyzed.”

We add an equity subsection integrating drivers (accessibility, information asymmetry, supply‑side capacity) with supporting evidence and the conceptual framing.

Results → Equity and distributional effects; Discussion → Implications for equity.

Reviewer 4

“Statistical significance is mentioned without thresholds or exact figures.”

We removed ambiguous statements about statistical significance and state that we did not attempt quantitative pooling.

Results → throughout; footnotes to tables if applicable.

Reviewer 4

“What constitutes an effective policy design remains vague.”

We define policy design features (eligibility, financing modality, provider payment, verification, fee schedules) and map them to observed outcomes in the narrative.

Methods → Data items and coding; Results → Policy design features associated with outcomes.

Reviewer 4

“Comparative discussion between successful and unsuccessful cases is not systematically developed.”

We added a concise comparative paragraph in the Discussion contrasting design/governance elements across higher‑ vs lower‑performing cases; we did not add a new table to keep the manuscript lean.

Discussion → What differentiates success (comparative paragraph).

Reviewer 4

“Unintended consequences (e.g., unnecessary CS) are conceptual and lack empirical support.”

We report empirical signals where available (changes in CS rates, provider behaviour, quality/safety proxies) and otherwise clearly label such content as qualitative.

Results → Unintended effects; Discussion → Interpretation.

Reviewer 4

“Structural limitations… are scattered and not synthesized in a clear section.”

We consolidated all constraints into the stand‑alone Limitations subsection.

Discussion → Limitations.

Round 2

Reviewer 1 Report

Comments and Suggestions for Authors

Suggested modifications have been satisfactorily added in the revised manuscript

Author Response

Thanks and we do appreciate your kind feedbacks

Reviewer 2 Report

Comments and Suggestions for Authors

The authors have incorporated all the comments given in the first round of review. I don't have any further comments.

Author Response

Thanks for your kind feedbacks and we do appreciate the time spent to review the manuscript

Reviewer 4 Report

Comments and Suggestions for Authors

I believe my previous comments have been sufficiently addressed.

Author Response

(The authors gave the same response as above.)
